# Graph Neural Network Acceleration via Matrix Dimension Reduction

## Abstract

Graph Neural Networks (GNNs) have become the de facto method for machine learning on graph data (e.g., social networks, protein structures, code ASTs), but they require significant time and resource to train. One alternative method is Graph Neural Tangent Kernel (GNTK), a kernel method that corresponds to infinitely wide multi-layer GNNs. GNTK's parameters can be solved directly in a single step, avoiding time-consuming gradient descent. Today, GNTK is the state-of-the-art method to achieve high training speed without compromising accuracy. Unfortunately, solving for the kernel and searching for parameters can still take hours to days on real-world graphs. The current computation of GNTK has running time $O(N^4)$, where $N$ is the number of nodes in the graph. This prevents GNTK from scaling to datasets that contain large graphs. Theoretically, we present two techniques to speed up GNTK training while preserving the generalization error: (1) We use a novel matrix decoupling method to reduce matrix dimensions during the kernel solving. This allows us to reduce the dominated computation bottleneck term from $O(N^4)$ to $O(N^3)$. (2) We apply sketching to further reduce the bottleneck term to $o(N^\omega)$, where $\omega \approx 2.373$ is the exponent of current matrix multiplication. Experimentally, we demonstrate that our approaches speed up kernel learning by up to $19\times$ on real-world benchmark datasets.

## 1 Introduction

Graph Neural Networks (GNNs) have quickly become the de facto method for machine learning on graph data. GNNs have delivered ground-breaking results in many important areas of AI, including social networking Yang et al. (2020a), bio-informatics Zitnik & Leskovec (2017); Yue et al. (2020), recommendation systems Ying et al. (2018), and autonomous driving Weng et al. (2020); Yang et al. (2020b). However, efficient GNNs training has become a major challenge with the relentless increase in the complexity of GNN models and dataset sizes, both in terms of the number of graphs in a dataset and the sizes of the graphs.

Recently, a new direction for fast GNN training is to use Graph Neural Tangent Kernel (GNTK). Solving for the kernel and searching for the parameters in GNTK is equivalent to using gradient descent to train an infinitely wide multi-layer GNN. GNTK is significantly faster than iterative gradient descent optimization because solving the parameters in GNTK is just a single-step kernel learning process. In addition, GNTK allows GNN training to scale with GNN model sizes because the training time grows only linearly with the complexity of GNN models. However, GNTK training can still take hours to days on typical GNN datasets today.

Our key observation is that, during the process of solving parameters in GNTK, most of the training time and resource is spent on multiplications of large matrices. Let $N$ be the maximum number of nodes in the graphs, these matrices can have sizes as large as $N^2 \times N^2$! This means a single matrix multiplication takes at least $N^4$ time, and it prevents GNTK from scaling to larger graphs. Thus, in order to speed up GNTK training, we need to reduce matrix dimensions.

**Our Contributions.** We present two techniques to speed up GNTK: (1) We use a novel matrix decoupling method to reduce matrix dimensions during the training without harming the calculation results. This reduces the dominated computation bottleneck term from $O(N^4)$ to $O(N^3)$. (2) We propose a sketching method to further reduce the bottleneck term to $o(N^\omega)$, where $\omega \approx 2.373$ is the exponent of current matrix multiplication.

We provide theoretical results that the resulting randomized GNTK still has a good generalization bound. In experiments, we evaluate our method on standard graph classification benchmarks. Our method improves GNTK training time by up to $19\times$ while maintaining the same level of accuracy.

## 2 BACKGROUND

**Notations.** For a positive integer $n$, we define $[n] := \{1, 2, \cdots, n\}$. For two integers $a \leq b$, we define $[a, b] := \{a, a + 1, \cdots, b\}$, and $(a, b) := \{a + 1, \cdots, b - 1\}$. Similarly we define $[a, b)$ and $(a, b]$. For a full rank square matrix $A$, we use $A^{-1}$ to denote its true inverse. We define the big O notation such that $f(n) = O(g(n))$ means there exists $n_0 \in \mathbb{N}_+$ and $M \in \mathbb{R}$ such that $f(n) \leq M \cdot g(n)$ for all $n \geq n_0$. For a matrix $A$, we use $\|A\|$ or $\|A\|_2$ to denote its spectral norm. We use $\|A\|_F$ to denote its Frobenius norm. We use $A^\top$ to denote the transpose of $A$. For a matrix $A$ and a vector $x$, we define $\|x\|_A := \sqrt{x^\top A x}$. We use $\phi$ to denote the ReLU activation function, i.e. $\phi(z) = \max\{z, 0\}$. For a function $f : \mathbb{R} \to \mathbb{R}$, we use $f'$ to denote the derivative of $f$.

**Graph neural network (GNN).** A GNN has $L$ levels of AGGREGATE operations, each followed by a COMBINE operation. A COMBINE operation has $R$ fully-connected layers with output dimension $m$, and uses ReLU as non-linearity. In the end, the GNN has a READOUT operation that corresponds to the pooling operation of normal neural networks.

Consider a graph $G = (V, E)$ with $|V| = N$. Each node $u \in V$ has a feature vector $h_u \in \mathbb{R}^d$.

In GNN we will use vectors $h^{(l,r)}$ such that $l$ denotes the number of levels, and $r$ denotes the number of hidden layers. The size is $h_u^{(1,0)} \in \mathbb{R}^d$, and $h_u^{(l,0)} \in \mathbb{R}^m$ for all $l \in [2 : L]$. We define the initial vector $h_u^{(0,R)} = h_u \in \mathbb{R}^d, \forall u \in U$.

For any $l \in [L]$, the **AGGREGATE operation** aggregates the information from last level:

$$h_u^{(l,0)} := c_u \cdot \sum_{v \in \mathcal{N}(u) \cup \{u\}} h_v^{(l-1,R)}.$$

where $c_u \in \mathbb{R}$ is a scaling parameter, and $\mathcal{N}(u)$ is the set of neighbor nodes of $u$. The **COMBINE operation** then uses $R$ fully-connected layers with ReLU activation: $\forall r \in [R]$,

$$h_u^{(l,r)} := (c_\phi/m)^{1/2} \cdot \phi(W^{(l,r)} \cdot h_u^{(l,r-1)}) \in \mathbb{R}^m,$$

where $c_\phi \in \mathbb{R}$ is a scaling parameter, $W^{(1,1)} \in \mathbb{R}^{m \times d}$, and $W^{(l,r)} \in \mathbb{R}^{m \times m}$ for all $(l, r) \in [L] \times [R] \backslash \{(1, 1)\}$. We let $W := \{W^{(l,r)}\}_{l \in [L], r \in [R]}$. Finally, the output of the GNN on graph $G = (V, E)$ is computed by a **READOUT operation**:

$$f_{\text{gnn}}(W, G) := \sum_{u \in V} h_u^{(L,R)} \in \mathbb{R}^m.$$

For more details see Appendix Section B.1.

**Neural tangent kernel.** We briefly review the neural tangent kernel definition. Let $\phi$ denote the non-linear activation function, e.g. $\phi(z) = \max\{z, 0\}$ is the ReLU activation function.

**Definition 2.1** (Neural tangent kernel Jacot et al. (2018))**.** *For any input two data points* $x, z \in \mathbb{R}^d$, *we define the kernel mapping* $\mathsf{K}_{\text{ntk}} : \mathbb{R}^d \times \mathbb{R}^d \to \mathbb{R}$

$$\mathsf{K}_{\text{ntk}}(x, z) := \int_{w \sim \mathcal{N}(0, I_d)} \phi'(w^\top x) \phi'(w^\top z) x^\top z \, \mathrm{d}w$$

*where $\mathcal{N}(0, I_d)$ is a d-dimensional multivariate Gaussian distribution, and $\phi'$ is the derivative of activation function $\phi$. If $\phi$ is ReLU, then $\phi'(t) = 1$ if $t \geq 0$ and $\phi'(t) = 0$ if $t < 0$. Given $x_1, x_2, \cdots, x_n \in \mathbb{R}^d$, we define kernel matrix $K \in \mathbb{R}^{n \times n}$ as follows: $K_{i,j} = \mathsf{K}(x_i, x_j)$.*

The lower bound on smallest eigenvalue of neural tangent kernel matrix $K$ (say $\lambda = \lambda_{\min}(K)$, see Du et al. (2019b); Arora et al. (2019a;b); Song & Yang (2019); Brand et al. (2020)) and separability of input data points $\{x_1, x_2, \cdots, x_n\}$ ($\delta$, see Li & Liang (2018); Allen-Zhu et al. (2019a;b)) play a crucial role in deep learning theory. Due to Oymak & Soltanolkotabi (2020), $\lambda$ is at least $\Omega(\delta/n^2)$ which unifies the two lines of research. The above work shows that as long as the neural network is sufficiently wide $m \geq \text{poly}(n, d, 1/\delta, 1/\lambda)$ (where $n$ is the number of input data points, $d$ is the dimension of each data) , running (S)GD type algorithm is able to minimize the training loss to zero.

**Kernel regression and equivalence.** Kernel method or Kernel regression is a fundamental tool in machine learning Avron et al. (2017; 2019); Scholkopf & Smola (2018). Recently, it has been shown that training an infinite-width neural network is equivalent to kernel regression Arora et al. (2019a). Further, the equivalence even holds for regularized neural network and kernel ridge regression Lee et al. (2020).

Let's consider the following neural network, $\min_W \frac{1}{2}\|Y - f_{\mathrm{nn}}(W, X)\|_2$. Training this neural network is equivalent to solving the following neural tangent kernel ridge regression problem: $\min_\beta \frac{1}{2}\|Y - f_{\mathrm{ntk}}(\beta, X)\|_2^2$. Note that $f_{\mathrm{ntk}}(\beta, x) = \Phi(x)^\top \beta \in \mathbb{R}$ and $f_{\mathrm{ntk}}(\beta, X) = [f_{\mathrm{ntk}}(\beta, x_1), \cdots, f_{\mathrm{ntk}}(\beta, x_n)]^\top \in \mathbb{R}^n$ are the test data predictors. Here, $\Phi$ is the feature map corresponding to the neural tangent kernel (NTK):

$$\mathsf{K}_{\mathrm{ntk}}(x, z) = \mathop{\mathbb{E}}_{W}\left[\left\langle \frac{\partial f_{\mathrm{nn}}(W, x)}{\partial W}, \frac{\partial f_{\mathrm{nn}}(W, z)}{\partial W} \right\rangle\right]$$

where $x, z \in \mathbb{R}^d$ are any input data, and $w_r \overset{i.i.d.}{\sim} \mathcal{N}(0, I)$, $r = 1, \cdots, m$.

## 3 OUR GNTK FORMULATION

We show our GNTK formulation in this section. Our formulation builds upon the GNTK formulas of Du et al. (2019a). The descriptions in this section is presented in a simplified way. See Section B.2 and B.3 for more details.

### 3.1 EXACT GNTK FORMULAS

We consider a GNN with $L$ AGGREGATE operations and $L$ COMBINE operations, and each COMBINE operation has $R$ fully-connected layers. Let $G = (U, E)$ and $H = (V, F)$ be two graphs with $|U| = N$ and $|V| = N'$. We use $A_G$ and $A_H$ to denote the adjacency matrix of $G$ and $H$. We give the recursive formula to compute the kernel value $\mathsf{K}_{\mathrm{gntk}}(G, H) \in \mathbb{R}$ induced by this GNN, which is defined as

$$\mathsf{K}_{\mathrm{gntk}}(G, H) := \mathop{\mathbb{E}}_{W \sim \mathcal{N}(0, I)}\left[\left\langle \frac{\partial f_{\mathrm{gnn}}(W, G)}{\partial W}, \frac{\partial f_{\mathrm{gnn}}(W, H)}{\partial W} \right\rangle\right],$$

where $\mathcal{N}(0, I)$ is a multivariate Gaussian distribution.

Recall that the GNN uses scaling factors $c_u$ for each node $u \in G$. We define $C_G$ to be the diagonal matrix such that $(C_G)_u = c_u$ for any $u \in U$. Similarly we define $C_H$. We will use intermediate matrices $K^{(\ell,r)}(G, H) \in \mathbb{R}^{N \times N'}$ for each $\ell \in [0 : L]$ and $r \in [0 : R]$, where $l$ denotes the level of AGGREGATE and COMBINE operations, and $r$ denotes the level of fully-connected layers inside a COMBINE operation.

Initially we define $K^{(0,R)}(G, H) \in \mathbb{R}^{N \times N'}$ as follows: $\forall u \in U, v \in V$,

$$[K^{(0,R)}(G, H)]_{u,v} := \langle h_u, h_v \rangle.$$

where $h_u, h_v \in \mathbb{R}^d$ are the input features of $u$ and $v$.

Next we recursively define $K^{(\ell,r)}(G, H)$ for $l \in [L]$ and $r \in [R]$.

| Reference | Time |
|---|---|
| Du et al. (2019a) | $O(n^2) \cdot (\mathcal{T}_{\mathrm{mat}}(N, N, d) + L \cdot N^4 + LR \cdot N^2)$ |
| Thm. 4.1 and 5.1 | $O(n^2) \cdot (\mathcal{T}_{\mathrm{mat}}(N, N, d) + L \cdot \mathcal{T}_{\mathrm{mat}}(N, N, b) + LR \cdot N^2)$ |

Table 1: When $L = O(1)$ and $R = O(1)$, the dominate term in previous work is $O(N^4)$. We improve it to $\mathcal{T}_{\mathrm{mat}}(N, N, b)$.

**Exact AGGREGATE.**   The AGGREGATE operation gives the following formula:

$$[K^{(\ell,0)}(G, H)]_{u,v} := c_u c_v \sum_{a \in \mathcal{N}(u) \cup \{u\}} \sum_{b \in \mathcal{N}(v) \cup \{v\}} [K^{(\ell-1,R)}(G, H)]_{a,b}.$$

In the experiments the above equation is computed using Kronecker product:

$$\mathrm{vec}(K^{(\ell,0)}(G, H)) := ((C_G A_G) \otimes (C_H A_H)) \cdot \mathrm{vec}(K^{(\ell-1,R)}(G, H)). \tag{1}$$

The dominating term of the final running time does not come from COMBINE and READOUT operations, thus we defer their details into Section B.2 in Appendix.

We briefly review the running time in previous work.

**Theorem 3.1** (Running time of Du et al. (2019a), simplified version of Theorem D.2)**. *Consider a GNN with $L$ AGGREGATE operations and $L$ COMBINE operations, and each COMBINE operation has $R$ fully-connected layers. We compute the kernel matrix using $n$ graphs $\{G_i = (V_i, E_i)\}_{i=1}^n$ with $|V_i| \leq N$. Let $d \in \mathbb{N}_+$ be the dimension of the feature vectors. The total running time is $O(n^2) \cdot (\mathcal{T}_{\mathrm{mat}}(N, N, d) + L \cdot N^4 + LR \cdot N^2)$.*

When using GNN, we usually use constant number of operations and fully-connected layers, i.e., $L = O(1), R = O(1)$, and we have $d = o(N)$, while the size of the graphs can grow arbitrarily large. Thus it is easy to see that the dominated term in the above running time is $N^4$, the major contribution of this work is to reduce it to $o(N^\omega)$, where $\omega \approx 2.373$ is the exponent of current matrix multiplication.

### 3.2 APPROXIMATE GNTK FORMULAS

We follow the notations of previous section. Now the goal is to compute an approximate version of the kernel value $\widetilde{K}(G, H) \in \mathbb{R}$ such that $\widetilde{\mathsf{K}}_{\mathrm{gntk}}(G, H) \approx \mathsf{K}_{\mathrm{gntk}}(G, H)$. We will use intermediate matrices $\widetilde{K}^{(\ell,r)}(G, H) \in \mathbb{R}^{N \times N'}$ for each $\ell \in [0 : L]$ and $r \in [0 : R]$. In the approximate version we add two random Gaussian matrices $S_G \in \mathbb{R}^{b \times N}$ and $S_H \in \mathbb{R}^{b' \times N'}$ in the AGGREGATE operation, where $b \leq N$ and $b' \leq N'$.

**Approximate AGGREGATE operation.**   In the approximate version, we add two sketching matrices $S_G \in \mathbb{R}^{b \times N}$ and $S_H \in \mathbb{R}^{b' \times N'}$:

$$\widetilde{K}^{(\ell,0)}(G, H) := C_G A_G \cdot (S_G^\top S_G) \cdot \widetilde{K}^{(\ell-1,R)}(G, H) \cdot (S_H^\top S_H) \cdot A_H C_H. \tag{2}$$

Not that for the special case $S_G^\top S_G = S_H^\top S_H = I$, the Eq. (2) degenerates to the the following case:

$$K^{(\ell,0)}(G, H) = C_G A_G \cdot K^{(\ell-1,R)}(G, H) \cdot A_H C_H.$$

This equation is exactly the same as the equation Eq. (1) of the exact case. See Fact 4.2 for why they are equivalent.

## 4 OUR TECHNIQUES : RUNNING TIME

The main contribution of our paper is to show that we can accelerate the computation of GNTK defined in Du et al. (2019a), while maintaining a similar generalization bound.

In this section we present the techniques that we use to achieve faster running time. We will provide the generalization bound in the next section. We first present our main running time theorem.

**Theorem 4.1** (Main theorem, running time part, Theorem D.1). *Consider a GNN with $L$ AGGREGATE operations and $L$ COMBINE operations, and each COMBINE operation has $R$ fully-connected layers. We compute the kernel matrix using $n$ graphs $n$ graphs $\{G_i = (V_i, E_i)\}_{i=1}^n$ with $|V_i| \leq N$. Let $b = o(N)$ be the sketch size. Let $d \in \mathbb{N}_+$ be the dimension of the feature vectors. The total running time is*

$$O(n^2) \cdot (\mathcal{T}_{\mathrm{mat}}(N, N, d) + L \cdot \mathcal{T}_{\mathrm{mat}}(N, N, b) + LR \cdot N^2).$$

Note that we improve the dominating term from $N^4$ to $\mathcal{T}_{\mathrm{mat}}(N, N, b)$. We achieve this improvement using two techniques:

1. We accelerate the multiplication of a Kronecker product with a vector by decoupling it into two matrix multiplications of smaller dimensions. In this way we improve the running time from $N^4$ down to $\mathcal{T}_{\mathrm{mat}}(N, N, N)$. We present more details in Section 4.2.

2. We further accelerate the two matrix multiplications by using two sketching matrices. In this way, we improve the running time from $\mathcal{T}_{\mathrm{mat}}(N, N, N)$ to $\mathcal{T}_{\mathrm{mat}}(N, N, b)$. We present more details in Section 4.3.

## 4.1    NOTATIONS AND KNOWN FACTS

**Fast matrix multiplication.**   We use the notation $\mathcal{T}_{\mathrm{mat}}(n, d, m)$ to denote the time of multiplying an $n \times d$ matrix with another $d \times m$ matrix. Let $\omega$ denote the exponent of matrix multiplication, i.e., $\mathcal{T}_{\mathrm{mat}}(n, n, n) = n^\omega$. The first result shows $\omega < 3$ is Strassen (1969). The current best exponent is $\omega \approx 2.373$, due to Williams (2012); Le Gall (2014). The common belief is $\omega \approx 2$ in the computational complexity community Cohn et al. (2005); Williams (2012); Jiang et al. (2020). The following fact is well-known in the fast matrix multiplication literature Coppersmith (1982); Strassen (1991); Bürgisser et al. (1997) : $\mathcal{T}_{\mathrm{mat}}(a, b, c) = O(\mathcal{T}_{\mathrm{mat}}(a, c, b)) = O(\mathcal{T}_{\mathrm{mat}}(c, a, b))$ for any positive integers $a, b, c$.

**Kronecker product and vectorization.**   Given two matrices $A \in \mathbb{R}^{n_1 \times d_1}$ and $B \in \mathbb{R}^{n_2 \times d_2}$. We use $\otimes$ to denote the Kronecker product, i.e., for $C = A \otimes B \in \mathbb{R}^{n_1 n_2 \times d_1 d_2}$, the $(i_1 + (i_2 - 1) \cdot n_1, j_1 + (j_2 - 1) \cdot d_1)$-th entry of $C$ is $A_{i_1, j_1} B_{i_2, j_2}$, $\forall i_1 \in [n_1], i_2 \in [n_2], j_1 \in [d_1], j_2 \in [d_2]$. For any give matrix $H \in \mathbb{R}^{d_1 \times d_2}$, we use $h = \mathrm{vec}(H) \in \mathbb{R}^{d_1 d_2}$ to denote the vector such that $h_{j_1 + (j_2 - 1) \cdot d_1} = H_{j_1, j_2}$, $\forall j_1 \in [d_1], j_2 \in [d_2]$.

## 4.2    SPEEDUP VIA KRONECKER PRODUCT EQUIVALENCE

We make the following observation about kronecker product and vectorization. Proof is delayed to Section E.

**Fact 4.2** (Equivalence between two matrix products and Kronecker product then matrix-vector multiplication). *Given matrices $A \in \mathbb{R}^{n_1 \times d_1}$, $B \in \mathbb{R}^{n_2 \times d_2}$, and $H \in \mathbb{R}^{d_1 \times d_2}$, we have $\mathrm{vec}(AHB^\top) = (A \otimes B) \cdot \mathrm{vec}(H)$.*

In GNTK, when computing the $l$-th AGGREGATE operation for $l \in [L]$, we need to compute a product $(A \otimes B) \cdot \mathrm{vec}(H)$ with sizes $A, B, H \in \mathbb{R}^{N \times N}$. Note that $A = C_G A_G, B = C_H A_H, H = K^{(l-1,R)}(G, H)$ (see Eq. (1) in Section 3.1). Computing this product naively takes $O(N^4)$ time, since the Kronecker product $A \otimes B$ already has size $N^2 \times N^2$. This is the $O(N^4)$ term of Du et al. (2019a).

In our experiments, we instead compute $AHB^\top$, which takes $O(\mathcal{T}_{\mathrm{mat}}(N, N, N))$ time. And this is how we get our first improvement in running time.

## 4.3    SPEEDUP VIA SKETCHING MATRICES

The following lemma shows that the sketched version of matrix multiplication approximates the exact matrix multiplication. This justifies why we can use sketching matrices to speed up calculation.

**Lemma 4.3** (Informal version of Lemma 5.4). *Given $n^2$ matrices $H_{1,1}, \cdots, H_{n,n} \in \mathbb{R}^{N \times N}$ and $n$ matrices $A_1, \cdots, A_n \in \mathbb{R}^{N \times N}$. Let $S_i \in \mathbb{R}^{b \times N}$ denote a random matrix where each entry is $+\frac{1}{\sqrt{b}}$ or $-\frac{1}{\sqrt{b}}$, each with probability $\frac{1}{2}$. Then with high probability, we have the following guarantee: for all $i, j \in [n]$,*

$$A_i^\top S_i^\top S_i H_{i,j} S_j^\top S_j A_j \approx A_i^\top H_{i,j} A_j.$$

Note that the sizes of the matrices are $A_i, A_j, H_{i,j} \in \mathbb{R}^{N \times N}$, and $S_i, S_j \in \mathbb{R}^{b \times N}$. They correspond to $A_i = C_{G_i} A_{G_i}, A_j = C_{G_j} A_{G_j}, H_{i,j} = \widetilde{K}^{(l-1,R)}(G_i, G_j)$ (see Eq. (2) in Section 3.2).

Directly computing $A_i^\top H_{i,j} A_j$ takes $\mathcal{T}_{\mathrm{mat}}(N, N, N)$ time.

After adding two sketching matrices, using a certain ordering of computation, we can avoid the time-consuming step of multiplying two $N \times N$ matrices. More specifically, we compute $A_i^\top S_i^\top S_i H_{i,j} S_j S_j^\top A_j$ in the following order:

- Computing $A_i^\top S_i^\top$ and $S_j A_j$ both takes $\mathcal{T}_{\mathrm{mat}}(N, N, b)$ time.
- Computing $S_i \cdot H_{i,j}$ takes $\mathcal{T}_{\mathrm{mat}}(b, N, N)$ time.
- Computing $(S_i H_{i,j}) \cdot S_j^\top$ takes $\mathcal{T}_{\mathrm{mat}}(b, N, b)$ time.
- Computing $(A_i^\top S_i^\top) \cdot (S_i H_{i,j} S_j)$ takes $\mathcal{T}_{\mathrm{mat}}(N, b, b)$ time.
- Computing $(A_i^\top S_i^\top S_i H_{i,j} S_j) \cdot (S_j A_j)$ takes $\mathcal{T}_{\mathrm{mat}}(N, b, N)$ time.

Thus, we improve the running time from $\mathcal{T}_{\mathrm{mat}}(N, N, N)$ to $\mathcal{T}_{\mathrm{mat}}(N, N, b)$.

## 5 OUR TECHNIQUES : ERROR ANALYSIS

In this section, we prove that even though adding the sketching matrices in the GNTK formula will introduce some error, this error can be bounded, and we can still prove a similar generalization bound as that of Du et al. (2019a).

**Theorem 5.1** (Informal version of Theorem C.4). *For each $i \in [n]$, if the labels $\{y_i\}_{i=1}^n$ satisfy $y_i = \alpha_1 \sum_{u \in V} \langle \overline{h}_u, \beta_1 \rangle + \sum_{l=1}^T \alpha_{2l} \sum_{u \in V} \langle \overline{h}_u, \beta_{2l} \rangle^{2l}$, where $\overline{h}_u = c_u \sum_{v \in \mathcal{N}(u) \cup \{u\}} h_v$, $\alpha_1, \alpha_2, \alpha_4, \cdots \alpha_{2T} \in \mathbb{R}$, $\beta_1, \beta_2, \beta_4, \cdots, \beta_{2T} \in \mathbb{R}^d$, and under the assumptions of Assumption C.6, and if we further have the conditions that*

$$4 \cdot \alpha_1 \|\beta_1\|_2 + \sum_{l=1}^T 4\sqrt{\pi}(2l-1) \cdot \alpha_{2l} \|\beta_{2l}\|_2 = o(n), \quad N = o(\sqrt{n}),$$

*then the generalization error of the approximate GNTK can be upper bounded by*

$$L_\mathcal{D}(f_{\mathrm{gntk}}) = \mathop{\mathbb{E}}_{(G,y) \sim \mathcal{D}}[\ell(f_{\mathrm{gntk}}(G), y)] \lesssim O(1/n^c), \text{ where constant } c \in (0, 1).$$

We use a standard generalization bound of kernel methods of Bartlett & Mendelson (2002) (Theorem C.1) which shows that in order to prove a generalization bound, it suffices to upper bound $\|y\|_{\widetilde{K}^{-1}}$ and $\mathrm{tr}[\widetilde{K}]$. We present our bound on $\|y\|_{\widetilde{K}^{-1}}$. The bound on $\mathrm{tr}[\widetilde{K}]$ is simpler. For the full version of the proofs, please see Section C.

**Lemma 5.2** (Informal version of bound on $\|y\|_{\widetilde{K}^{-1}}$). *Under Assumption C.6, we have*

$$\|y\|_{\widetilde{K}^{-1}} \le 4|\alpha_1| \cdot \|\beta_1\|_2 + \sum_{l=1}^T 4\sqrt{\pi}(2l-1)|\alpha_{2l}| \cdot \|\beta_{2l}\|_2^{2l}.$$

We provide a high-level proof sketch here. We first compute all the variables in the approximate GNTK formula to get a close-form formula of $\widetilde{K}$. Then combining with the assumption on the labels $y$, we show that $\|y\|_{\widetilde{K}_1^{-1}}$ is upper bounded by

$$\|y\|_{\widetilde{K}_1^{-1}} \le (4\alpha^2 \cdot \beta^\top \overline{H} \cdot (\widetilde{H}^\top \widetilde{H})^{-1} \cdot \overline{H}^\top \beta)^{1/2},$$

where $\overline{H}, \widetilde{H} \in \mathbb{R}^{d \times n}$ are two matrices such that $\forall i, j \in [n]$,

$$[\overline{H}^\top \overline{H}]_{i,j} = \mathbf{1}_{N_i}^\top C_{G_i}^\top A_{G_i}^\top H_{G_i}^\top \cdot H_{G_j} A_{G_j} C_{G_j} \mathbf{1}_{N_j},$$

$$[\widetilde{H}^\top \widetilde{H}]_{i,j} = \mathbf{1}_{N_i}^\top C_{G_i}^\top A_{G_i}^\top (S_{G_i}^\top S_{G_i}) H_{G_i}^\top \cdot H_{G_j} (S_{G_j}^\top S_{G_j}) A_{G_j} C_{G_j} \mathbf{1}_{N_j}.$$

Next we show that $\widetilde{H}^\top \widetilde{H}$ is a PSD approximation of $\overline{H}^\top \overline{H}$.

**Claim 5.3** (PSD approximation). *We have* $(1 - \frac{1}{10})\overline{H}^\top \overline{H} \preceq \widetilde{H}^\top \widetilde{H} \preceq (1 + \frac{1}{10})\overline{H}^\top \overline{H}$.

Note that using this claim, we have

$$\|y\|_{\widetilde{K}_1^{-1}} \leq (4\alpha^2 \cdot \beta^\top \overline{H} \cdot (\widetilde{H}^\top \widetilde{H})^{-1} \cdot \overline{H}^\top \beta)^{1/2} \leq (8\alpha^2 \cdot \beta^\top \overline{H} \cdot (\overline{H}^\top \overline{H})^{-1} \cdot \overline{H}^\top \beta)^{1/2} \leq 4 \cdot \alpha \|\beta\|_2,$$

which finishes the proof. Now it remains to prove the claim. We prove it by using the following lemma which upper bounds the error of adding two sketching matrices. The proof of this lemma is deferred to Section E.

**Lemma 5.4** (Error bound of adding two sketching matrices). *Let* $R \in \mathbb{R}^{b_1 \times n}, S \in \mathbb{R}^{b_2 \times n}$ *be two independent AMS matrices* Alon et al. (1999). *Let* $\beta = O(\log^{1.5} n)$. *For any matrix* $A \in \mathbb{R}^{n \times n}$ *and any vectors* $g, h \in \mathbb{R}^n$, *the following holds with probability* $1 - 1/\text{poly}(n)$

$$|g^\top R^\top RAS^\top Sh - g^\top Ah| \leq \frac{\beta}{\sqrt{b_1}} \|g\|_2 \|Ah\|_2 + \frac{\beta}{\sqrt{b_2}} \|g^\top A\|_2 \|h\|_2 + \frac{\beta^2}{\sqrt{b_1 b_2}} \|g\|_2 \|h\|_2 \|A\|_F.$$

Using this lemma and the assumption of sketching sizes in the lemma statement, we can prove the following coordinate-wise upper bound:

$$|[\widetilde{H}^\top \widetilde{H}]_{i,j} - [\overline{H}^\top \overline{H}]_{i,j}| \leq \frac{1}{10} \cdot [\overline{H}^\top \overline{H}]_{i,j}.$$

Then we can upper bound $\|\widetilde{H}^\top \widetilde{H} - \overline{H}^\top \overline{H}\|_2 \leq \frac{1}{10} \|\overline{H}^\top \overline{H}\|_2$, which proves the claim. Thus we finish the proof.

## 6 EXPERIMENTS

In this section, we evaluate our proposed GNTK acceleration algorithm on various graph classification tasks. More details about the experiment setup can be found in Section F of the supplementary material.

**Datasets.** We test our method on 7 benchmark graph classification datasets, including 3 social networking dataset (COLLAB, IMDBBINARY, IMDBMULTI) and 4 bioinformatics datasets (PTC, NCL1, MUTAG and PROTEINS) Yanardag & Vishwanathan (2015). For bioinformatics dataset, each node has its categorical features as input feature $h$ to the algorithm. For social network dataset where nodes have no input feature, we use degree of each node as its feature to better represent its structural information. The dataset statistics are shown in Table 2.

**Baselines.** We compare our proposed results with a number of state-of-the-art baselines for graph classification: (1) State-of-the-art deep graph neural networks architectures, including Graph Convolution Network (GCN) Kipf & Welling (2017), GraphSAGE Hamilton et al. (2017), PATCHY-SAN Niepert et al. (2016), Deep Graph CNN (DGCNN) Zhang et al. (2018a) and Graph Isomorphism Network (GIN) Xu et al. (2018a). (2) Kernel based methods, including the WL subtree kernel Shervashidze et al. (2011), Anonymous Walk Embeddings (AWL) Ivanov & Burnaev (2018), and RetGK Zhang et al. (2018b). (3) Graph neural tangent kernel (GNTK) Du et al. (2019a). For deep learning methods, GNTK, RetGK and AWL, we report accuracy reported in the original papers. For WL subtree, we report the accuracy of the implementation used in Xu et al. (2018a).

**Results.** We perform 10-fold cross validation and report the mean and standard deviation of accuracy. We show our performance by comparing with state-of-the-art Graph learning methods, including the original GNTK method. The accuracy is shown in Table 2 and

| Datasets | COLLAB | IMDB-B | IMDB-M | PTC | NCI1 | MUTAG | PROTEINS |
|---|---|---|---|---|---|---|---|
| # of graphs | 5000 | 1000 | 1500 | 344 | 4110 | 188 | 1113 |
| # of classes | 3 | 2 | 3 | 2 | 2 | 2 | 2 |
| Avg # of nodes | 74.5 | 19.8 | 13.0 | 25.5 | 29.8 | 17.9 | 39.1 |
| GCN | $79.0 \pm 1.8$ | $74.0 \pm 3.4$ | $51.9 \pm 3.8$ | $64.2 \pm 4.3$ | $80.2 \pm 2.0$ | $85.6 \pm 5.8$ | $76.0 \pm 3.2$ |
| GraphSAGE | $-$ | $72.3 \pm 5.3$ | $50.9 \pm 2.2$ | $63.9 \pm 7.7$ | $77.7 \pm 1.5$ | $85.1 \pm 7.6$ | $75.9 \pm 3.2$ |
| PATCHY-SAN | $72.6 \pm 2.2$ | $71.0 \pm 2.2$ | $45.2 \pm 2.8$ | $60.0 \pm 4.8$ | $78.6 \pm 1.9$ | $92.6 \pm 4.2$ | $75.9 \pm 2.8$ |
| DGCNN | $73.7$ | $70.0$ | $47.8$ | $58.6$ | $74.4$ | $85.8$ | $75.5$ |
| GIN | $80.2 \pm 1.9$ | $75.1 \pm 5.1$ | $52.3 \pm 2.8$ | $64.6 \pm 7.0$ | $82.7 \pm 1.7$ | $89.4 \pm 5.6$ | $76.2 \pm 2.8$ |
| WL Subtree | $78.9 \pm 1.9$ | $73.8 \pm 3.9$ | $50.9 \pm 3.8$ | $59.9 \pm 4.3$ | $86.0 \pm 1.8$ | $90.4 \pm 5.7$ | $75.0 \pm 3.1$ |
| AWL | $73.9 \pm 1.9$ | $74.5 \pm 5.9$ | $51.5 \pm 3.6$ | $-$ | $-$ | $87.9 \pm 9.8$ | $-$ |
| RetGK | $81.0 \pm 0.3$ | $71.9 \pm 1.0$ | $47.7 \pm 0.3$ | $62.5 \pm 1.6$ | $84.5 \pm 0.2$ | $90.3 \pm 1.1$ | $75.8 \pm 0.6$ |
| GNTK | $83.6 \pm 1.0$ | $76.9 \pm 3.6$ | $52.8 \pm 4.6$ | $67.9 \pm 6.9$ | $84.2 \pm 1.5$ | $90.0 \pm 8.5$ | $75.6 \pm 4.2$ |
| **Ours** | $83.6 \pm 1.0$ | $76.9 \pm 3.6$ | $52.8 \pm 4.6$ | $67.9 \pm 6.9$ | $84.2 \pm 1.5$ | $90.0 \pm 8.5$ | $75.6 \pm 4.2$ |

Table 2: **Classification accuracy (%) for graph classification datasets with matrix decoupling**. We report the result of our proposed method, optimizing on original GNTK model.

| Datasets | COLLAB | IMDB-B | IMDB-M | PTC | NCI1 | MUTAG | PROTEINS |
|---|---|---|---|---|---|---|---|
| GNTK | $> 24$ hrs | $546.4$ | $686.0$ | $46.5$ | $10,084.7$ | $8.0$ | $1,392.0$ |
| **Ours** | $4,523.0\ (> \mathbf{19\times})$ | $90.7\ (\mathbf{6\times})$ | $112.5\ (\mathbf{6.1\times})$ | $13.5\ (\mathbf{3.4\times})$ | $7,446.8\ (\mathbf{1.4\times})$ | $3.0\ (\mathbf{2.7\times})$ | $782.7\ (\mathbf{1.8\times})$ |

Table 3: **Running time analysis for our matrix decoupling method (in seconds)**. We report the kernel calculation time between the original GNTK method and our accelerated model.

running time is shown in Table 3. Our matrix decoupling method (MD) doesn't harm the result of GNTK while significantly accelerates the learning time of neural tangent kernel. Our proposed method achieves multiple times of improvements for all the datasets. In particular, on COLLAB, our method achieves more than 19 times of learning time acceleration. We observe that the improvement of our method depends on the sizes of the graphs. For large-scale dataset like COLLAB, we achieve highest acceleration because matrix multiplication dominates the overall calculation time. And for bioinformatics datasets where number of nodes is relatively small, the improvement is not as prominent. Note that we only show the running time comparison between our method and the original GNTK method, because other state-of-the-art deep GNN methods takes significantly longer to learn via gradient descent Du et al. (2019a). Analysis of our sketching method can be found in Section F of the supplementary material.

## 7    CONCLUSION

Graph Neural Networks (GNNs) have become the most important method for machine learning on graph data (e.g., social networks, protein structures), but training GNNs efficiently is a major challenge. One alternative method is Graph Neural Tangent Kernel (GNTK), a kernel method that is equivalent to train infinitely wide multi-layer GNNs using gradient descent. GNTK's parameters can be solved directly in a single step, avoiding time-consuming gradient descent. Because of this, GNTK has become the state-of-the-art method to achieve high training speed without compromising accuracy. Unfortunately, GNTK still takes hours to days to train on real-world graphs because it has a computation bottleneck of $O(N^4)$, where $N$ denotes the number of nodes in the graph. We present two techniques to mitigate this bottleneck: (1) We use a novel matrix decoupling method to reduce matrix dimensions during the kernel solving. This allows us to reduce this dominated computation bottleneck from $O(N^4)$ to $O(N^3)$. (2) We apply sketching to further reduce the bottleneck to $o(N^\omega)$, where $\omega \approx 2.373$ is the exponent of current matrix multiplication. We demonstrate that our approaches speed up kernel learning by up to $19\times$ on real-world benchmark datasets.

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
