# OpenReview forum: "Graph Neural Network Acceleration via Matrix Dimension Reduction"
_ICLR.cc/2021/Conference — Reject_

### Official Review · AnonReviewer2 · 2020-10-27
**Good idea but lacks novelty**

**Rating:** 5
**Confidence:** 3

**Review:**

This paper proposes to apply Neural Tangent Kernel to GNN to improve training. This is a good initiative to apply state of the art methods to GNN training which would interest researchers and practitioners in the community. However, the novelty introduced in the paper low since the authors directly apply NTK to GNN.

Pros:
a.) A good point of the paper is the compressive experiments and theoretical guarantees which make the paper complete and useful to the community. The paper is also well written.

b.) Good performances have been achieved from experiments with many benchmark datasets


Cons:
a.) My main issue with the paper is its lack of novelty. It is a good idea but the authors do not provide any new ideas for either NTK and GNN. The paper shows significant performance improvement with the proposed method, however, it is somewhat understandable.

b.) More details about the experiments would be helpful. Can the authors give details about varying dimensions of the sketch matrices and its effect on speed and accuracy?

c.) This paper focuses on using sketch matrices based on Alon et al.. It would be more informative if the authors compare using other sketching methods such as Gaussian matrices and sampling methods.

---

> ### Author Response · Authors · 2020-11-23
> **Response to AnonReviewer2 (Part 1/2)**
>
> Thanks for the detailed feedback. We address the issues below.
>
> - My main issue with the paper is its lack of novelty.
>
> Answer: This paper proposed two techniques to speed up GNTK computation: matrix decoupling, and sketching in GNTK, and we believe they are novel and will be useful for future research.
>
> We especially want to stress the novelty of our design of combining the sketching methods with GNTK by comparing it with previous research. In the standard “sketch and solve” paradigm, sketching methods are used for numerical linear algebra problems, e.g., linear regression, low rank approximation (see [1] and [2]). Recently sketching methods are also used for iterative methods (including the iterative optimization algorithm for deep learning), and they can be used with or without precomputation. In [3] the algorithm precomputes the product of multiple sketching matrices with a fixed matrix, and uses a different copy in each iteration. In [4] there is no precomputation, and the algorithm sketches a matrix on the fly.
>
> We remark that these previous results all use sketching methods in the following way: 1. They always add sketching matrices on one side of a matrix to accelerate the computation of the matrix. 2. They use sketching methods to preserve subspace embedding and approximate matrix product.
>
> However, in our paper, we add sketching matrices on both sides of a matrix, and we add sketching matrices when computing each entry of the matrix. This is a totally new scenario and requires totally new ideas.
>
> We also want to remark that the previous papers that use sketching in deep learning are usually not practical, e.g., [4] accelerates second order methods that are only good in theory but not in practice. Our sketching technique not only has theoretical improvement but we believe it will have experimental improvement when the datasets are large enough.
>
> - More details about the experiments would be helpful. Can the authors give details about varying dimensions of the sketch matrices and its effect on speed and accuracy?
>
> Answer: In the supplementary section F, we provide more implementation and experiment details. And we also provide some evidence to study how matrix sketching affects accuracy and time.
>
> Since currently the graph datasets are not large enough to demonstrate the effectiveness of our sketching method, we only consider a toy problem that corresponds to our sketching error lemma (Lemma 5.4): Following Lemma 5.4, we validate the running time and error difference between matrix multiplication with and without the sketching method. Specifically, we randomly generate [n, n] matrix $A$, $G$ and $H$. And matrix multiplication without sketching is calculated by $G^T A H$. For the sketching method, we randomly generate two AMS matrices $R$ and $S$ with size $[\gamma n, n]$ where $\gamma$ is the sketching ratio. And matrix multiplication with sketching is calculated by $G^T R^T R A S^T S H$. For error, we run experiments under different sketching rates from $0.1$ to $0.5$. Experiments show that our sketching error is always lower than the theoretical bound, and using sketching results in a shorter running time. We also observe that when sketching rate gets higher, the error decreases and in the meantime running time increases because the dimension of the matrix is larger, and we lose less information. This validates our Lemma 5.4, showing that our matrix sketching method has a strictly bounded error. The figure of sketching error and running time comparison are added in section F of our supplementary file.
>
> Existing benchmark graph classification datasets [5][6] only provides graphs with average node number no more than 500. With these datasets, we show a 19x speedup with our matrix decoupling method, however, the limited size (< 500 nodes per graph) of the datasets do not allow us to study the end-to-end accuracy performance trade-off.  Our paper studies the theoretical aspects of both our matrix decoupling method and sketching method extensively but only provides empirical results for matrix decoupling method.  Although we have proved error bound for our matrix sketching method, the implication on end-to-end performance and accuracy on real large graphs is still an open question.

---

> ### Author Response · Authors · 2020-11-23
> **Response to AnonReviewer2 (Part 2/2)**
>
> - It would be more informative if the authors compare using other sketching methods such as Gaussian matrices and sampling methods.
>
> Answer: Our theoretical analysis also holds for other common dense sketching matrices, including Gaussian matrices and SRHT matrices. But we cannot use sparse embedding matrices or count-sketch matrices ([2]). This is because the error in the main sketching lemma (Lemma 5.4) becomes too large with these sparse matrices.
>
> Leverage score sampling [7][8] can also be used in many places to replace sketching matrices, e.g,, subspace embedding and approximate matrix product. For a matrix A, the leverage score of the $i$-th row $a_i^{\top}$ of $A$ is defined as $\tau_i = a_i^{\top} (A^{\top} A)^{-1} a_i$. Leverage score sampling then samples each row of A with probability $\tau_i$. Note leverage score sampling is not oblivious since it depends on the matrix. Also, it is not known how to do it on both sides of a matrix. Thus at this point we do not know if leverage score sampling can be used in our application of GNTK.
>
> [1] Kenneth L. Clarkson, David P. Woodruff. Low-rank PSD approximation in input-sparsity time. SODA 2017.
>
> [2] Jelani Nelson, Huy L. Nguyên. OSNAP: Faster numerical linear algebra algorithms via sparser subspace embeddings. FOCS 2013.
>
> [3] Yin Tat Lee, Zhao Song, Qiuyi Zhang. Solving Empirical Risk Minimization in the Current Matrix Multiplication Time. COLT 2019.
>
> [4] Jan van den Brand, Binghui Peng, Zhao Song, and Omri Weinstein. Training (Overparametrized) Neural Networks in Near-Linear Time. ITCS 2021.
>
> [5] Yanardag, Pinar, and S. V. N. Vishwanathan. Deep graph kernels. ACM SIGKDD 2015.
>
> [6] Hu, Weihua, Matthias Fey, Marinka Zitnik, Yuxiao Dong, Hongyu Ren, Bowen Liu, Michele Catasta, and Jure Leskovec. Open graph benchmark: Datasets for machine learning on graphs. arXiv preprint arXiv:2005.00687 (2020).
>
> [7]  Daniel A. Spielman, Nikhil Srivastava. Graph sparsification by effective resistances. SIAM Journal on Computing 40, no. 6 (2011): 1913-1926.
>
> [8] Joshua Batson, Daniel A. Spielman, and Nikhil Srivastava. Twice-ramanujan sparsifiers. SIAM Journal on Computing 41, no. 6 (2012): 1704-1721.

---

### Official Review · AnonReviewer1 · 2020-10-27
**A good paper on accelerating graph neural tangent kernel**

**Rating:** 5
**Confidence:** 1

**Review:**

In this paper, the authors propose two techniques to speed up the training of the graph neural tangent kernel, by matrix decoupling and sketching. Experiments are convincing.

We find the title of the paper inappropriate, because the paper only consider the  graph neural tangent kernel.

In general, the paper is difficult to read because many important parts are only available in the appendices.

There are some spelling and grammatical errors that can be easily identified and corrected, such as "The descriptions in this section is".

----------------
Following the authors' response,  we have updated our rating accordingly considering the following facts: the authors didn't make any change to the paper within the rebuttal, while they had the possibility in response to several questions. They didn't address even our simplest concerns, about the title being inappropriate. They still want to keep the title too general, while the paper considers only graph neural tangent kernel. Even if the latter is equivalent to infinitely wide multi-layer GNN, it is still a very special case. Moreover, many of the major issues raised by the other reviews were not addressed.

---

> ### Author Response · Authors · 2020-11-23
> **Response to AnonReviewer1**
>
> We thank the reviewer for the positive feedback.
>
> - We find the title of the paper inappropriate, because the paper only consider the graph neural tangent kernel.
>
> Answer: As explained in Section 2 background and related work, GNTK is equivalent to infinitely wide multi-layer GNNs and GNTK currently is the one of the state-of-the-art methods to train GNNs in terms of achieving accuracy. That’s why we use GNN in the title. We are open to more discussion for different options for the title.
>
> - In general, the paper is difficult to read because many important parts are only available in the appendices.
>
> Answer: We put all the main results and a proof sketch (with a list of crucial lemmas for the proof) in the first ten pages, for example, our main generalization bound is presented in Theorem 5.1, and the key lemma for proving this theorem is Lemma 5.2, which then uses the sketching bound Lemma 5.4. Unfortunately, due to the lack of space, we have to delay the full proof to the appendices. We will further improve the writing to make it clearer when revising the paper.
>
> - There are some spelling and grammatical errors that can be easily identified and corrected, such as "The descriptions in this section is".
>
> Answer: We will fix the typos and grammar errors when revising the paper.

---

### Official Review · AnonReviewer4 · 2020-10-28
**Interesting but misses convincing evaluation and has questionable results.**

**Rating:** 4
**Confidence:** 2

**Review:**


#### **Post Rebuttal**
I'm afraid the authors failed to answer my main question regarding the results and the applicability of their proposed approximation.

Therefore, I decide to keep my score.

---
#### **Contribution Claims**
The paper vouches for the use of GNTK as a strong method for graph learning combining good properties of kernels and GNNs. However, it raises the issue of time complexity for calculating the kernel scaling with $O(N^4)$, and addresses it by suggesting two techniques to gain speed up. The proposed techniques are based on matrix decoupling, which simply improves complexity, and matrix sketching which approximates the correct kernel (in probability).
The authors further present a theoretical result that similar generalization guarantees, as attributed to the exact GNTK, are achieved with their approximation.

**Strengths**
- The paper gains a major speed up in GNTK calculation with no harm to performance (*up to a clarification regarding the reported results. See first item in weaknesses section)
- The method generalization properties are analyzed and shown to be preserved.
- Although the paper is packed with lots of details, the flow of the paper allows the reader to pick upon the general ideas presented in the paper.

**Weaknesses**
- *Results* - I have concerns regarding the reported performance in Table 2. I find it quite uncanny that the numbers are *exactly* the same as achieved by exact GNTK, although the proposed method is an approximation which also utilizes a stochastic component. In case I misunderstood, I would kindly ask the authors to clarify this point.
- *Discussion compared to GNTK* - A discussion on how the exact GNTK generalization bound compares to the approximate one is missing.
- *Related work* - The paper does not explain and mention previous and relevant uses of matrix sketching (either in deep learning or kernel methods in general) which makes it hard for someone who is not familiar with the term to understand solely from the paper what it means. It is also unclear how innovative the use of it is.
- *Experimental setting* - Regarding the evaluation on social datasets, it is mentioned that a degree feature was added as an input. Is it true for the other baselines as well?
- *Missing convincing use-cases* - As the paper claims to gain speed up making GNTK scalable to larger datasets, I would have liked to see the performance both in time and accuracy of the proposed method.

#### **Decision Recommendation**
For now, I suggest to reject the paper.
I think that it is interesting and valuable but feels that it needs to be solidified.
Upon clarification of the questionable results I would consider changing my decision.

#### **Other Comments**
- Typos - the paper contains many typos. e.g, Theorem 4.1 "...n graphs n graphs..." repeating twice.

---

> ### Author Response · Authors · 2020-11-23
> **Response to AnonReviewer4 (Part 1/2)**
>
> We thank the reviewer for the constructive comments. We would like to clarify the concerns raised by the reviewer.
>
> - Concerns regarding the reported performance in Table 2: I find it quite uncanny that the numbers are exactly the same as achieved by exact GNTK.
>
> Answer: As stated in the caption, Table 2 shows the results of our matrix decoupling method. In Section 4.2 of the paper, we show that our proposed matrix decoupling method is equivalent to the original Kronecker product while significantly reducing the computation time. Thus, the accuracy should not be affected at all with our matrix decoupling method. The results shown in Table 2 are expected.
>
>
> - Discussion compared to GNTK - A discussion on how the exact GNTK generalization bound compares to the approximate one is missing.
>
> Answer: Compared to the exact GNTK generalization bound, our approximate one uses different assumptions on the data and labels. More specifically, we assume the labels to be the form of finite summation (see part 1 of Assumption C.6), while in the exact case they allowed the labels to be the form of infinite summation. Also, we add two more reasonable assumptions about the feature vectors and sketching sizes (see part 2 and 3 of Assumption C.6).
> We will add more discussion about this comparison in the revised version.
>
>
> - The paper does not explain and mention previous and relevant uses of matrix sketching (either in deep learning or kernel methods in general).
>
> Answer: Sketching method has been extensively applied in kernel methods, e.g. [1][2][3][4]. It has also been used in theoretical deep learning, e.g. [5]. We compare our design of combining the sketching methods with GNTK with previous research.
>
> In the standard “sketch and solve” paradigm, sketching methods are used for numerical linear algebra problems, e.g., linear regression, low rank approximation (see [10] and [11]). Recently sketching methods are also used for iterative methods (including the iterative optimization algorithm for deep learning), and they can be used with or without precomputation. In [12] the algorithm precomputes the product of multiple sketching matrices with a fixed matrix, and uses a different copy in each iteration. In [5] there is no precomputation, and the algorithm sketches a matrix on the fly.
>
> We remark that these previous results all use sketching methods in the following way: 1. They always add sketching matrices on one side of a matrix to accelerate the computation of the matrix. 2. They use sketching methods to preserve subspace embedding and approximate matrix product.
>
> However, in our paper, we add sketching matrices on both sides of a matrix, and we add sketching matrices when computing each entry of the matrix. This is a totally new scenario and requires totally new ideas.
> We also want to remark that the previous papers that use sketching in deep learning are usually not practical, e.g., [5] accelerates second order methods that are only good in theory but not in practice. Our sketching technique not only has theoretical improvement but we believe it will have experimental improvement when the graphs in the datasets are large enough.
>
> - Experimental setting - Regarding the evaluation on social datasets, it is mentioned that a degree feature was added as an input. Is it true for the other baselines as well?
>
> Answer: As stated in “Dataset” part, the nodes of bioinformatics dataset obtain categorical features initially, and we use that feature as the input h to the network. In comparison, nodes in the social network dataset do not have any pre-defined feature, so we treat the degree of each node as its feature h and input to the network. This is standard practice, also done in [6][7].

---

> > ### Comment · AnonReviewer4 · 2020-11-24
> > **Some more questions**
> >
> > I thank the authors for their answers. I do have some more comments / questions:
> >
> > - **Reported performance** - I now understand my confusion about the reported results, I thought the method used was also incorporated with the matrix sketching, which has is an approximation and not an equivalence.
> > However, in that case, I think it is very good the authors showed the speed up gained by matrix decomposition, but what happens to performance when sketching is used?
> >
> > - **Experiments showing the sketching error** - Thank you for adding that. Yet, same question, how does it effect the performance of the computed kernel?

---

> > > ### Author Response · Authors · 2020-11-24
> > > **Response to Follow-up Comments**
> > >
> > > Thanks for reading our response.  Our paper studies the theoretical aspects of our matrix decoupling method and our sketching method. However, we only provide empirical evaluations for our matrix decoupling method on real-world datasets. The speedup is up to 19x. On those real-world datasets (< 500 nodes per graph), after our matrix decoupling method is applied, matrix multiplication is no longer the bottleneck for GNTK training. Because we do not know any public graph datasets that have larger graphs, we didn't evaluate our sketching method empirically in terms of the performance-accuracy trade-off. We believe, however, as we are applying GNN to larger and larger graphs in the future for larger social networks or more complex protein structures, our sketching method will be useful, and this paper provides the theoretical understanding of the generalization error bound for the sketching method. The reviewer is correct that the implication on end-to-end performance accuracy trade-off on real-world large graphs for our sketching method is still an open question.

---

> ### Author Response · Authors · 2020-11-23
> **Response to AnonReviewer4 (Part 2/2)**
>
> - As the paper claims to gain speed up making GNTK scalable to larger datasets, I would have liked to see the performance both in time and accuracy of the proposed method.
>
> Answer: The largest open graph datasets we can find are the ones presented in the existing benchmark graph classification datasets [8][9]. These datasets only contain graphs of average no more than 500 nodes. However, we envision that the sizes of graphs will grow in the future as graph neural networks are used for larger social networks and more complex protein structures.
>
> We demonstrate a 19x speedup on these real-world datasets using our matrix decoupling method without loss of accuracy. Unfortunately, the size of these datasets (< 500 nodes per graph) is not large enough for us to study the end-to-end performance-accuracy tradeoff for our sketch method. With a small number of nodes per graph, the overall running time is not dominated by matrix multiplication after our matrix decoupling method is applied, so the acceleration is not significant. Our paper studies the theoretical aspects of both our matrix decoupling method and sketching method extensively but only provides empirical results for matrix decoupling method.  Although we have proved error bound for our matrix sketching method, the implication on end-to-end performance and accuracy on real large graphs is still an open and interesting question.
>
> To this end, we provide some evidence to show how matrix sketching affects accuracy and time : Following Lemma 5.4, we validate the running time and error difference between matrix multiplication with and without the sketching method. Specifically, we randomly generate $[n, n]$ matrix $A$, $G$ and $H$. And matrix multiplication without sketching is calculated by $G^T A H$. For the sketching method, we randomly generate two AMS matrices $R$ and $S$ with size $[\gamma n, n]$ where $\gamma$ is the sketching ratio. And matrix multiplication with sketching is calculated by $G^T R^T R A S^T S H$. For error, we run experiments under different sketching rates from $0.1$ to $0.5$. Experiments show that our sketching error is always lower than the theoretical bound, and using sketching results in a shorter running time. We also observe that when sketching rate gets higher, the error decreases and in the meantime running time increases because the dimension of the matrix is larger, and we lose less information. This validates our Lemma 5.4, showing that our matrix sketching method has a strictly bounded error. The figure of sketching error and running time comparison are added in section F of our supplementary file.
>
> **References**:
>
> [1] Jason D. Lee, Ruoqi Shen, Zhao Song, Mengdi Wang, Zheng Yu. Generalized Leverage Score Sampling for Neural Networks. NeurIPS 2020.
>
> [2] Haim Avron, Michael Kapralov, Cameron Musco, Christopher Musco, Ameya Velingker, Amir Zandieh. Random Fourier features for kernel ridge regression : approximation bounds and statistical guarantees. ICML 2017
>
> [3] Haim Avron, Michael Kapralov, Cameron Musco, Christopher Musco, Ameya Velingker, Amir Zandieh. A universal sampling method for reconstructing signals with simple Fourier transform. STOC 2019
>
> [4] Josh Alman, Timothy Chu, Aaron Schild, Zhao Song. Algorithms and Hardness for Linear Algebra on Geometric Graphs. FOCS 2020
>
> [5] Jan van den Brand, Binghui Peng, Zhao Song, and Omri Weinstein. Training (Overparametrized) Neural Networks in Near-Linear Time. ITCS 2021.
>
> [6] Du, Simon S., Kangcheng Hou, Russ R. Salakhutdinov, Barnabas Poczos, Ruosong Wang, and Keyulu Xu. Graph neural tangent kernel: Fusing graph neural networks with graph kernels. NeurIPS 2019.
>
> [7] Xu, Keyulu, Weihua Hu, Jure Leskovec, and Stefanie Jegelka. How powerful are graph neural networks?. ICLR 2019.
>
> [8] Yanardag, Pinar, and S. V. N. Vishwanathan. Deep graph kernels. ACM SIGKDD 2015.
>
> [9] Hu, Weihua, Matthias Fey, Marinka Zitnik, Yuxiao Dong, Hongyu Ren, Bowen Liu, Michele Catasta, and Jure Leskovec. Open graph benchmark: Datasets for machine learning on graphs. arXiv preprint arXiv:2005.00687 (2020).
>
> [10] Kenneth L. Clarkson, David P. Woodruff. Low-rank PSD approximation in input-sparsity time. SODA 2017.
>
> [11] Jelani Nelson, Huy L. Nguyên. OSNAP: Faster numerical linear algebra algorithms via sparser subspace embeddings. FOCS 2013.
>
> [12] Yin Tat Lee, Zhao Song, Qiuyi Zhang. Solving Empirical Risk Minimization in the Current Matrix Multiplication Time. COLT 2019.

---

### Decision · Program_Chairs · 2021-01-07
**Final Decision**

**Decision:**

Reject

**Comment:**

While the author response clarified some concerns, it could not convince the reviewers that the current version of the paper should be accepted for publication at ICLR.